# GPX2 Gene Affects Feed Efficiency of Pigs by Inhibiting Fat Deposition and Promoting Muscle Development

**DOI:** 10.3390/ani12243528

**Published:** 2022-12-14

**Authors:** Lei Pu, Yunyan Luo, Zuochen Wen, Yuxin Dai, Chunting Zheng, Xueli Zhu, Lei Qin, Chunguang Zhang, Hong Liang, Jianbin Zhang, Liang Guo, Lixian Wang

**Affiliations:** 1Tianjin Key Laboratory of Agricultural Animal Breeding and Healthy Husbandry, College of Animal Science and Veterinary Medicine, Tianjin Agricultural University, Tianjin 300384, China; 2Institute of Animal Science, Chinese Academy of Agricultural Sciences, Beijing 100193, China

**Keywords:** GPX2, SNP, backfat, feed efficiency, molecular mechanism, adipocyte, muscle cells, development

## Abstract

**Simple Summary:**

Glutathione peroxidase 2 (GPX2) gene expression differs between individuals with different GPX2 polymorphism. Individuals with high GPX2 expression had thinning 100 kg backfat (BF) thickness, no change on average daily gain (AG) and lower feed conversion ratio (FCR) and residual feed intake (RFI) in Duroc pigs. The reason for this result could be that increasing GPX2 could inhibit the adipocyte proliferation and differentiation, promoting the lipid degradation. Meanwhile, GPX2 may promote muscle cells’ proliferation and myogenic differentiation. Overall, our fundings indicate that GPX2 can transfer more nutrients to less fat deposition direction, providing the material basis for animal weight gain.

**Abstract:**

GPX2 has been recognized as a potential candidate gene for feed efficiency in pigs. This article aimed to elucidate polymorphism of GPX2 associated with feed efficiency and its related molecular mechanism. In this study, seven single nucleotide polymorphisms (SNP) of GPX2 were found among 383 Duroc pigs. In addition, seven SNPs and ALGA0043483 (PorcineSNP60 BeadChip data in 600 Duroc pigs), which are near the GPX2 gene, were identified in one haplotypes block. Furthermore, associated studies showed that the genotype of GPX2 has significant association with weaning weight and 100 kg BF in Duroc pigs. In addition, the AG had no effect when the backfat became thinner, and the FCR and RFI traits had a tendency to decrease in the G3 + TT combination genotype, accompanied by an increase of GPX2 expression in backfat and muscle tissues. At the cellular level, the adipocyte proliferation and ability of adipogenic differentiation were reduced, and the lipid degradation increased in 3T3-L1 when there was overexpression of GPX2. In contrast, overexpression of the GPX2 gene can promote the muscle cell proliferation and myogenic differentiation in C2C12 cells. In other words, GPX2 has the effect of reducing fat deposition and promoting muscle development, and it is a candidate gene for backfat and feed efficiency.

## 1. Introduction

Glutathione peroxidase 2 (GPX2) could play an important role in protecting mammals from toxicity of hydroperoxides and reduce lipid peroxides [1]. GPX2 is involved in the regulation of lipid peroxidation. It may serve as a marker gene for inflammation and intestinal homeostasis [2]. Loss of epithelium-specific GPX2 results in aberrant cell fate decisions during intestinal differentiation, and played the principal role in intestinal epithelium proliferation and apoptosis [3,4]. The expression of GPX2 was different in Angus longissimus dorsi at different growth stages [5]. GPX2 played an important role in cell proliferation and differentiation in lung adenocarcinoma cells [6]. Ob/ob mice showed a decrease in adipocyte cellular activity of GPXs [7]. Inhibition of GPX2 activity impaired insulin signaling in 3T3-L1 adipocytes and was involved in the insulin resistance of obesity [7].

A genome-wide association study (GWAS) suggested that one significant SNP nearest to GPX2 was significantly associated with feed efficiency of pigs and deduced that GPX2 plays an major role in RFI [8]. In addition, quantitative trait loci (QTL) in Porcine chromosome 7 was found to correlate with backfat on the 3–4th lumbar vertebrae, 11–12th rib and neck fat thickness [9]. Other similar reports have found the backfat traits were affected by the QTL (region: 75–117 cm on porcine chromosome 7), which contained GPX2 [10]. In addition, several studies have shown that the GPX2 gene was involved in lipid metabolism [11]. Therefore, GPX2 is considered to be an important candidate gene for feed utilization efficiency and porcine fat deposition. Taken together, GPX2 performs fundamental functions in protecting cells from the toxicity of hydroperoxides and lipid peroxides, cell growth and development, which processes were highly influential for feed efficiency traits. However, there was not any similar investigation on the association of the GPX2 gene with feed efficiency traits. The present study was conducted to detect sequence variants of GPX2 in Duroc pigs (Sus scrofa domestics), investigate their associations with feed efficiency traits and explore their influence mechanism in the cellular level.

## 2. Materials and Methods

All animals were treated following the Council of China guidelines for the care of experimental animals. All procedures in this study were performed according to protocols approved by the Tianjin key laboratory of agricultural animal breeding and healthy husbandry, College of Animal Science and Veterinary Medicine, Tianjin Agricultural University, (Tianjin, China).

### 2.1. Animals and Data Collection

The diet of the pigs met or exceeded the National Research Council (NRC) (1998) requirements [12]. Body weight (BW) was recorded in birth and weaning. Six hundred Duroc pigs of similar ages (approximately 80 days old) were gradually placed in the same enclosure. Electronic Feed Intake Recording Equipment (FIRE, Osborne, KS, USA) was utilized to obtain daily feed intake (DFI) and body weight during the test period. Experiments started on approximately 90-day-old pigs (on-test) and proceeded until body weight reached approximately 180 days old (off-test). The 10th rib backfat (BF) was measured using an Aloka ultrasound machine (Corometrics Medical Systems, Inc., Wallingford, CT, USA) when the pigs reached approximately 100 kg in body weight. AG was calculated as the slope of a simple linear regression of BW for the number of days during the test. This study was based on 223 boars and 377 gilts. The predicted equation for RFI was carried out by a linear regression model that was previously described [13] as follows:RFI = DFI − [b1 × onBW + b2 × offBW + b3 × metamidBW + b4 × ADGA + b5 × offBFA + e]
where onBW was the body weight on-test, offBW was the body weight off-test, metamidBW was the metabolic mid body weight (average weight raised to the 0.75 power), ADGA was adjusted AG to testing from 90 to 180 days old, and offBFA was the off-test BF adjusted to 100 kg of BW. The regression coefficients (b1 to b5) and regression intercept (e) were computed using a multiple linear regression model. The formula for the calculations at age 30 and 100 kg was as described by the Canadian swine improvement program (2010) [14]. FCR was determined as the ratio of mass gain to feed intake.

### 2.2. Genomic DNA and RNA Isolation

Genomic DNA for the genotypic was extracted from ear samples by the phenol-chloroform procedure. DNA samples were stored at −20 °C [14]. Heart, liver, spleen, lungs, kidney, muscle, backfat, pancreas, ileum and stomach tissues were harvested from three Duroc pigs of 100 kg, flash frozen in liquid nitrogen and stored at −80 °C.

Total RNA was extracted from these tissue samples using the traditional TRIzol^®^ Method. The cDNA was cloned using the PrimeScriptTM RT reagent kit (TaKaRa, Tokyo, Japan), according to the manufacturer instructions. Detected the genotypic of GPX2 SNP.

Thirty individuals were randomly sampled to identify GPX2 SNPs. The mRNA and genomic sequences of GPX2 were obtained from the GenBank database (http://www.ncbi.nlm.nih.gov/genbank/) (accessed on 10 July 2020) (accessions DQ898282.2 and FP236589.2, respectively). Six pairs of primers (Appendix A) were designed to amplify the 5′ flanking regions and exons of GPX2 by polymerase chain reaction (PCR). PCR amplification was performed in a 50 μL volume containing 2 μL DNA sample (DNA concentration: 100–200 ng/μL), 10 μM of forward and reverse primers, 250 μM dNTPs, 1 U of Taq polymerase in 5 μL 10 × polymerase Buffer (TaKaRa, Japan), and H_2_O up to the 50 μL total volume. PCR conditions were as follows: an initial denaturation stage at 94 °C for 5 min, 36 cycles of denaturation at 94 °C for 30 s, annealing at 53.5 °C for 40 s and extension at 72 °C for 50 s, and a final extension stage at 72 °C for 10 min. The products were scored on a 1.5% agarose gel after GoldView staining and sequenced at SinoGenoMax Co., Ltd. (Beijing, China). Polymorphic sites were analyzed using the DNAstar software [15]. Genotyping was performed for 383 Duroc pigs using matrix-assisted laser desorption, ionization-time-of-flight (MALDI-TOF) mass spectrometry (QIAGEN, Hilden, Germany). Primers and probes for genotypic assays for seven SNPs of GPX2 are shown in Table 1 and Appendix A.

A total of six hundred Duroc pigs were genotyped using the PorcineSNP60 BeadChip (Illumina, San Diego, CA, USA). The genotypic platform used in this research was an Infinium II Multi-Sample Assay. SNP chips were scanned using iScan and analyzed using Illumina GenomeStudio (Illumina, Inc., 9885 Towne Centre Drive, San Diego, CA, USA). To assess the technical reliability of genotyping, two or more randomly selected DNA samples were genotyped, and over 99.9% identity was obtained.

### 2.3. Secondary Structure Predicted by RNAsnp

RNAsnp (http://rth.dk/resources/rnasnp/) (accessed on 12 July 2020) was used to predict the effect of SNPs on the local RNA secondary structure. RNA secondary structures were established by hydrogen bonding, which formed the double strand and ring structure. The base pair probabilities of the wild-type and mutant RNA sequences were calculated using the global folding method, RNAfold. The structural difference between wild-type and mutants was computed using Euclidean distance or Pearson correlation measures for all sequence intervals (or local regions).

### 2.4. 3T3-L1 Cell Culture and Adipogenic Differentiation

The 3T3-L1 cells were removed from the liquid nitrogen tank and immediately put into a 38 °C water bath for recovery. Dulbecco minimum essential medium (DMEM) containing 10% fetal bovine serum (FBS) and 100 units/mL penicillin–streptomycin was mixed well in pipette, inoculated into 60 mm cell culture dishes and cultured in a 37 °C, 5% CO_2_ incubator. The 3T3-L1 cells reached 80% confluence, trypsin was added to the cells for 3–5 min, an equal amount of medium to stop the digestion, transferred to a centrifuge tube, centrifuged for 5 min, removed the supernatant, added the medium, and mixed by pipetting. Cells were seeded at a ratio of 1:3, passaged, and the medium was altered every day. After 1 day of cell contact inhibition, adipogenic induction started. The 3T3-L1 cells were treated with an induction medium containing 0.5 mg/L dexamethasone, 0.5 mmol/L IBMX and 10 µg/mL insulin to induce adipogenic differentiation for 2 days, and then maintained in a culture medium containing only 10 µg/mL insulin. The medium was cultured for 4 d, followed by subsequent treatment.

### 2.5. C2C12 Cell Culture and Myogenic Differentiation

The method of cell resuscitation and subculture was as of 3T3-L1. The cells were cultured in DMEM containing 10% FBS and 100 units/mL penicillin–streptomycin. Cell cultures were conducted at 37 °C in 5% CO_2_ incubator. When 60–70% confluence was reached, trypsin was used to digest and for passage of the cells. After the cells converged, DMEM contained 4% horse serum and 100 units/mL penicillin–streptomycin was used for myogenic differentiation, and the cells were collected after 6 days of induction.

### 2.6. GPX2 Adenovirus Overexpressed

Construction of the GPX2 adenovirus overexpression vector (pAV-EF1A > {Sus scrofa GPX2(ns)}:P2A:EGFP). HEK293A cells were utilized to amplify the virus to achieve >10^10^ IFU/mL Virus titer of concentration. Then, 2 μL GPX2 adenovirus was added into the cell passage (30 mm in diameter), in the process of detecting proliferation effect. We added 2 μL GPX2 adenovirus when cells grew to 70% density (30 mm in diameter), when the cells grow to a certain density, then induced further differentiation. The effect of GPX2 overexpression on cell differentiation was detected.

### 2.7. EdU Proliferation Staining

After the cells were passaged, the cells were cultured for 24 h for EdU proliferation detection. We pre-warmed 400 μL of 50 uM EdU working solution at 37 °C, added it to the Petri dish and continued to culture in the incubator for 2 h. After EdU labeling the cells, the culture medium was removed, 500 μL of 4% paraformaldehyde was added and fixed for 30 min at room temperature. The fixative was removed and the cells were washed 2 times with PBS for 3–5 min each. We added 50 μL 2 mg/mL Glycine solutions, and shaking table incubated at room temperature for 5 min, the cells were washed 1–2 times with PBS for 3–5 min each time. Then, 400 μL of penetrant (0.5% Trition X-100 in PBS) was added, and shaking table incubated at room temperature for 10 min, and the cells were washed 1 time with PBS 3–5 min. In the dark 400 μL Apollo 567 stains was added and shaking table incubated at room temperature for 30 min, the Apollo medium was removed, 2 times with 400 μL of penetrant (0.5% Trition X-100 in PBS), shaking table incubated at room temperature for 10 min 1–2 times with 400 μL of methanol, it was washed for 5 min, 1 time PBS for 5 min. Hoechest 33342 was added for nuclear staining for 30 min, Hoechest 33342 was removed, and the cells were washed 1–3 times with PBS for 3 min each. Observed with a fluorescence microscope in the dark and took pictures.

### 2.8. Oil Red O Staining and Quantification

The cells were cultured at the terminal differentiation stage, and the adipocytes were washed 3 times with PBS, 4% cell fixation solution was added, and the cells were fixed in 37 °C incubators for 45 min. After washing three times with PBS, Oil red O solution was added for 45 min. The staining of lipid droplets was observed under a microscope, and pictures were taken. The Oil red O was dissolved with 1 mL isopropanol for colorimetry (60 mm diameter cell culture dish), and the absorbance of the sample was analyzed with a microplate reader at 510 nm wavelength.

### 2.9. Immunofluorescence Staining

We added 2 μL GPX2 adenovirus when cells grew to 70% density. After the cells converged, DMEM containing 4% horse serum and 100 units/mL penicillin–streptomycin was used for myogenic differentiation, and the cells were collected after 6 days of induction. Cells were fixed with 4% paraformaldehyde at room temperature for 15 min and permeability with 0.1% Triton X-100 for 10 min. After blocking for 30 min on ice, cell nuclei were stained by incubating the cells in 1 mg/mL DAPI for 5 min at room temperature. Fluorescence was observed with a Motic fluorescence microscope (Motic, Xiamen, China). Since the virus overexpression vector contains green fluorescent protein, which fills the muscle fiber, the muscle differentiation rate and the myotube fusion index were calculated directly after nuclear staining.

### 2.10. Quantitative Real-Time PCR (qRT-PCR)

Glyceraldehyde-3-phosphate dehydrogenase (GAPDH) was used as the reference gene to normalize the expression levels in the different tissues (heart, liver, spleen, lungs, kidney, muscle, backfat, pancreas, ileum and stomach tissues). The PCR reaction used 1 μL of cDNA (Concentration 100–200 ng/μL), 7.5 μL SYBR Select Master Mix (ABI, Los Angeles, CA, USA), 5.5 μL RNase-free H_2_O, and 0.5 μL of the forward and reverse primers (listed in Appendix A). Each qRT-PCR cycle was conducted as follows: 50 °C for 2 min; 95 °C for 5 min; 36 cycles at 95 °C for 10 s and 60 °C for 1 min. Each independent experiment was conducted in triplicate. The 2^−ΔΔCT^ (CT: cycle threshold) method was used to calculate the relative gene expression levels [6]. The expression of genes was detected by qRT-PCR with the same methods. The primer sequences involved are in Appendix A.

### 2.11. Western Blotting Analysis

The cell was added to lysis buffer containing protease inhibitors and phosphatase inhibitors in the culture dish, lysis on ice for 3–5 min, and the cell solution was collected in a 1.5 mL centrifuge tube. Set aside on ice for 10 min, centrifuge at 12,000 r for 10 min at 4 °C, and collect the supernatant. An appropriate amount of 5 × protein loading buffer was added, mixed by gently pipetting, boiled in hot water at 100 °C for 10 min, and stored the protein samples at −80 °C. For Western blot detection, protein samples and markers were separated by SDS-PAGE gel electrophoresis and then transferred to PVDF membranes (Millipore, Bedford, MA, USA). After membrane transfer, the PVDF membrane was rinsed with TBST buffer for 10 min, blocked in 5% skim milk for 2 h, washed 3 times with TBST buffer, incubated with primary antibody overnight (4 °C) and then washed with TBST for 3 times. Second, the secondary antibody was incubated for 1.5 h, and washed three times with TBST. Diluted primary antibody according to the concentration specified in the instructions. The dilution ratio of the secondary antibody horseradish peroxidase was 1:20,000. ECL chemiluminescence method was used for color development, and finally, the target protein was detected by a gel imaging system, and the protein bands were analyzed in grayscale by Image J software.

### 2.12. Statistical Analysis

Allele and genotype frequencies were obtained for each polymorphism. Allele frequencies, genotype frequencies, He: gene heterozygosity. Ne: effective number of alleles. PIC: polymorphism information content. HWE: Hardy–Weinberg equilibrium values were obtained by GenAlEx 6.0. software (http://biologyassets.anu.edu.au/GenAlEx/Download.html) (accessed on 8 September 2021). The genetic haplotype definition was defined as several genes or SNPs that were statistically associated with each other and formed a gene locus or common patterns of genetic variation. Haploview 4.2 (http://www.broadinstitute.org/mpg/haploview) (accessed on 8 September 2021) was used to detect the haplotype block in all the SNPs of GPX2 gene and ALGA0043483. The least-squares method, applied in the generalized linear mixed model (GLMM), allowed for evaluating the association between feed efficiency traits and genotypes in SAS (SAS Inst. Inc., Cary, NC, USA). GLMM was as follows:Y = μ + G + S + P + A + W + e
where Y is the observation of the trait (Birth Weight, Weaning Weight, 90 dBW, 30 kg age), μ is the population mean, G is the fixed effect of the combination genotype, S is the random effect of sex—it did not include P and A as the random effect—W is the covariate effect of Birth Weight when Y was Weaning Weight), W is the covariate effect of Weaning Weight when Y was 90 dBW, 30 kg age)—when we analyzed Y as the Birth Weight, no covariable was set—and e is the random error.

Differential analysis of gene expression and protein expression were used in the independent sample of *t*-test method.

## 3. Results

### 3.1. GPX2 Polymorphisms and Genotype Frequencies and Haplotype Determination

The statistical description of the phenotypic data is presented in Appendix A. Six pairs of primers were designed to amplify the 5′ flanking regions and exons of GPX2 (Figure 1). After sequencing the GPX2, seven SNPs were revealed in 383 Duroc pigs. The results showed four SNPs within 5′ flanking regions (g.-774 G > T, g.-1043 T > C, g.-1333 A > G, and g.-1531 C > G), two SNPs within exons (c.182 C > T, c.665 T > C) and one SNP within a 3′-untranslated region (UTR) (c.1032 G > A). The c.182 C > T and c.665 T > C SNPs corresponded with synonymous mutations. We found that ALGA0043483, which was genotyped by the PorcineSNP60 BeadChip, was located 0.1 Mb upstream of GPX2 (it was known from the chromosome information in version 10.2, but not in version 11.1).

Detailed information about genotype and allele frequencies of the seven SNPs and ALGA0043483 is shown in Table 2. Allele frequencies of A, C, T, G, T, A, C and C in c.1032 G > A, c.665 T > C, c.182 C > T, g.-774 T > G, g.-1043 C > T, g.-1333 G > A, g.-1531 G > C and ALGA0043483, respectively, varied from 51.17% to 54.83% (Table 2).

The heterozygosity (He), effective number of alleles (Ne), polymorphism information content (PIC) and Hardy–Weinberg equilibrium value of seven SNPs and ALGA0043483 in 600 Duroc pigs was detected. The SNPs and ALGA0043483 have medium He, effective Ne and polymorphism information content (medium level: 0.25 < PIC < 0.5). Seven SNPs and ALGA0043483 present Hardy–Weinberg equilibrium (*p* < 0.05) in 600 Duroc pigs.

The schematic illustration of the SNP locations in the GPX2 gene, exon1 included the 5′UTR, and exon 2 contained the 3′UTR. Blue blocks represent the exons; black lines and blue lines in the exon region represent the 5′UTR and 3′UTR. Red blocks represent the 5′ flanking region. Amplification products of the 5′ flanking region and exons for the GPX2 gene were detected by the PCR technique. M: D2000 DNA marker; 5′ F1, 5′ F2, 5′ F3: different 5′ flanking region; E1: exon 1, E2-1: the upper section of exon 2; E2-2: lower half of exon 2.

This study found that ALGA0043483 had a high linkage with seven SNPs of GPX2 (D ≥ 0.98). Based on seven SNPs and ALGA0043483, only one haplotype block was detected in Duroc pigs (Figure 2). Five linkage genotypes (G1 to G5) comprised the haplotype block, and G1 to G3 were the dominant genotypes, with frequencies of 18.47%, 27.71%, and 53.03%, respectively (Table 3).

### 3.2. Prediction of the SNPs’ Effect on the Local RNA Secondary Structure

This study predicted the effect of three SNPs (c.1032 G > A, c.665 T > C, c.182 C > T) on the RNA secondary structure by RNAsnp software [16]. The results confirmed that the SNP of c.182 C > T and c.665 T > C did not lead to a great change in the GPX2 RNA secondary structure. In c.1032 G > A, the mutation of nucleotide A resulted in a significant RNA secondary structure difference in GPX2. The variation of c.1032 G > A could change minimum free energy −346.60 kcal/mol to −338.60 kcal/mol (Appendix A). Therefore, the variation of this SNP could change minimum free energy 8 kcal/mol. Compared to the RNA secondary structure in the wild and mutant genotypes on 944–1042 bp, we found that the stem–loop structure had completely changed from screw to lattice diagram (Appendix A).

### 3.3. Genotype and ALGA0043483 Association with Feed Efficiency Traits Expression Analysed in Different Tissues

An association analysis of 383 Duroc pigs is shown in Table 4. It confirmed that individuals with combination Genotype 3 (G3) showed significantly thinner backfat than combination Genotype 1 (G1) and combination Genotype 2 (G2) (*p* < 0.1). Unexpectedly, this study found that seven SNPs of GPX2 did not have a significant effect among birth weight, weaning weight, body weight at 90 days of age, ADFI, AG, 30 kg age, 100 kg age, FCR and RFI traits (Table 4). The relationship of ALGA0043483 genotypes with feed efficiency traits in 600 Duroc pigs. Similarly, the variation of ALGA0043483, did not have a significant effect on birth weight, body weight at 90 days of age, ADFI, AG, 30 kg age, 100 kg age, FCR and RFI traits (Table 5). In ALGA0043483 locus, statistical results showed that individuals with genotype TT had thinner (100 kg) backfat than CC and TC individuals (*p* < 0.01). Additionally, pigs with genotype TT appeared to have a higher weaning weight than CC and TC individuals (*p* < 0.05).

The RNA expression of GPX2 was highly expressed in the lungs and ileum (Figure 3). These results showed that GPX2 was more highly expressed in muscle (*p* < 0.05), backfat (*p* < 0.01) and ileum (*p* < 0.05) in G3 individuals than G1 individuals (Figure 4). There was a lower GPX2 expression in G3 individuals’ pancreas tissue (*p* < 0.05) than G1 individuals. There was no significant expression difference between G1 and G3 individuals in heart, liver, spleen, lungs, kidney and stomach.

### 3.4. Overexpression of GPX2 May Inhibit the Proliferation of 3T3-L1

Adenovirus overexpression of GPX2 was performed on 3T3-L1 cells to elucidate the effect of proliferation situation. This research found that GPX2 mRNA and protein were overexpressed in 3T3-L1 cells (Figure 4A). After overexpression of GPX2, the proliferation of 3T3-L1 cells was detected by EdU staining. Overexpression of GPX2 could significantly (*p* < 0.01) reduce the proliferation rate (34%) of 3T3-L1 cells (Figure 4B). The mRNA and protein expressions of CDK2 were decreased after GPX2 overexpression (Figure 4C,D). The expression of the P21 gene was increased, which is a marker gene associated with inhibition of cell proliferation. These results showed that overexpression of GPX2 gene could reduce the proliferation ability of 3T3-L1 cells

### 3.5. GPX2 Promoted Lipid Degradation and Inhibit Adipogenic Differentiation of 3T3-L1

Adipogenesis was assessed at day 7, and we found the lipid content of 3T3-L1 cell decreased in overexpression GPX2 than in control, as assessed by oil red O staining (Figure 5A). Quantification of lipid accumulation indicated that differentiation of 3T3-L1 was very significantly inhibited by overexpression of GPX2 (Figure 5B). The expression levels of adipogenesis and the lipid degradation related gene were analyzed (Figure 5C). The results showed that the mRNA expression levels of PPARγ dropped, and protein expression levels also reduced but not at a significant level (Figure 5D). The mRNA and protein level of CEBPα did not obviously change. However, the expression of AP2 was upregulated. Interestingly, the level of HSL and LPL genes, related to lipid degradation, was increased in overexpression of GPX2 than in control. Together the results suggest that the differentiation of 3T3-L1 cell was inhibited and enhanced lipid degradation by overexpression of GPX2.

### 3.6. Overexpression GPX2 Induces the Proliferation of C2C12

Adenovirus overexpression of GPX2 in C2C12 cells was used to access specificity in proliferation situation. GPX2 mRNA and protein were found to be overexpressed in C2C12 cells (Figure 6A). The proliferation of C2C12 cells was detected by EdU staining after GPX2 overexpression. Overexpression of GPX2 significantly (*p* < 0.01) increased the proliferation rate (41%) of C2C12 cells (Figure 6B). After GPX2 overexpression, the mRNA and protein expressions ofCDK2 and CyclinB increased (Figure 6C,D). It had no effect on the expression of the P21 gene, which is a marker gene associated with inhibition of cell proliferation. These findings indicated that overexpression of the GPX2 gene could boost C2C12 cell proliferation.

### 3.7. Overexpression GPX2 Promoted the Myoblastic Differentiation of C2C12

To determine the effect of GPX2 on myoblastic differentiation, C2C12 cells were overexpressed for GPX2, and then myoblast differentiation was induced. The cell contour was displayed using GFP in an adenovirus overexpression vector, and the nucleus was stained with DAPI. The C2C12 cells overexpressing GPX2 had no significant effect on the differentiation index compared to the control group, but significantly increased the fusion index (*p* < 0.05) (Figure 7A). MyoG, an early marker of myoblast differentiation, was elevated (Figure 7B,C). However, it had no discernible effect on the MyoD gene. The MYH3 alate marker of skeletal muscle differentiation increased significantly (*p* < 0.01) when GPX2 was overexpressed compared to the control group. The level of MSTN mRNA, which is a marker of the muscle growth suppressor gene, had no change.

## 4. Discussion and Conclusions

Onteru et al. (2013) discovered two SNPs 0.2–0.4 Mb upstream of the GPX2 gene that have been shown to be associated with RFI in Large White pigs [7]. Nonetheless, no systematic study of the association of GPX2 polymorphisms with RFI traits was conducted. In this present study, seven SNPs were found (four in the 5’ flanking region, and three in the exons). One of the seven SNPs (c.1032 G > A) changed the minimum free energy of the GPX2 secondary RNA structure by 8 kcal/mol. The SNP (c.1032 G > A) was then thought to play an important role in the secondary structure of GPX2. According to previous research, minimum free energy structures are a collection of suboptimal structures with similar free energy [14]. A secondary RNA structure was found to be important in transcription initiation, transcription, termination, RNA stability and gene expression [17]. Several correlational studies have hypothesized that the primary structure and the secondary structure of RNA play a crucial role in gene activity [18]. Based on past research, we hypothesized that GPX2 function may be influenced by a number of secondary structures This aspect needs further study.

Near GPX2, one SNP (ALGA0043483 in porcine chromosome 7) was considered to be strongly connected to seven SNPs of GPX2. Seven SNPs and ALGA0043483 corresponded to three major genotypes (G1, G2, G3). In 383 Duroc pigs, G3 individuals had considerably lower backfat than G1 and G2 individuals. In 600 Durocs, the TT genotype of ALGA0043483 individuals had thinner backfat than the CC and CT genotypes. The GPX2 gene was found to be substantially associated with backfat thickness in Durocs. Other research has yielded similar results. The quantitative trait loci (QTL) found on Porcine chromosome 7 have been linked to 3–4th lumbar vertebrae backfat, 11–12th rib backfat, backfat thickness and neck fat thickness [9]. Ponsuksili et al. (2005) also found that a QTL region (75–117 cM on porcine chromosome 7) that contained GPX2 affected backfat traits [10]. Similar studies detected that GPX2 was also involved in regulation of lipid peroxidation and perfluorooctanoic acid metabolism [11]. Numerous studies have shown that GPX2 could be a novel target in the Wnt pathway [19,20]. The promoter of GPX2 was activated by β-catenin and had a negative effect on lipogenesis [20,21]. In the present study, sequence variants association analysis showed there were no significant associations detected between genotypes with FCR and RFI. However, G3 individuals showed smaller FCR and RFI than G1 individuals. Interestingly, we found that individuals with thinner backfat had no decrease in daily weight gain, which could mean that a decrease in fat content accompanied by an increase in muscle content ultimately does not affect the daily weight gain trait (Figure 8).

Individuals with combination genotype G3 have much higher expression of GPX2 in muscle, backfat and ileum than G1 combination genotype individuals. In other words, the expression of GPX2 was higher in adipose tissue, muscle tissue and the small intestine of pigs with thinner backfat and more feed efficiency. To investigate the underlying mechanism, we first examined the effect of GPX2 on proliferation and adipogenic differentiation in 3T3-L1 cell. Further research revealed that overexpression of GPX2 could inhibit proliferation of 3T3-L1, inhibit adipogenic differentiation and promote adipose degradation. White adipocytes require pyruvate oxidation through the dehydrogenase system to produce acetyl-CoA citrate synthesis; pyruvate dehydrogenase-dependent reactive oxygen species production would be directly linked to de novo adipogenesis and triglyceride deposition [22]. Therefore, the process of lipid accumulation in adipocytes is accompanied by oxidative stress. It is speculated that overexpression of GPX2 can significantly alleviate oxidative stress in 3T3-L1 adipocytes. These results are in agreement with previous publications that reported selenium directly increased GPX1 expression in adipocytes and decreased adipocyte differentiation and lipid deposition at the same time [23]. The GPX2 and GPX1 genes have similar gene functions, but the expression levels of the two genes are different [7]. It was recently published that 3T3-L1 cells treated with the antioxidant component Tyrosol(TR) also showed that adipogenesis was inhibited by downregulated C/EBPα and PPARγ and a reduction of lipid droplets. Moreover, TR treatment also acted on the early stage of differentiation by reducing cell proliferation (~40%), as shown by the increase in p21 protein expression [24]. Myricetin significantly promoted GPX activity in high-fat diet(HFD)-fed mice, improved antioxidant capacity in vivo, and downregulated the expressions of adipogenic PPARγ and CEBPα genes in HFD-induced mice [25]. The addition of antioxidant lycopene can increase liver HSL and ATGL and promote liver lipolysis [26]. Another antioxidant had similar results in other animals or cells, reduced lipid synthesis and promoted lipolysis [27,28]. In AP2-dTg mice fed the high fat and sugar diet, the activities of antioxidant related enzymes were higher than wild-type mice. Meanwhile, Tissue hydrogen peroxide was significantly reduced in subcutaneous and gonadal White adipose tissue (WAT) of obese AP2-dTg mice [29]. The expression of AP2 is inversely related to oxidative stress. The higher the expression of AP2, the lower the degree of oxidative stress. In the current study, we confirmed that overexpression of GPX2 could protect the cells from oxidative stress. The expression of AP2 was increased in mRNA and protein expression, which is consistent with the above research. Collectively, these results suggest adipogenesis is inhibited and lipolysis is promoted by overexpression of GPX2, and also reveal novel mechanism knowledge about GPX2 in adipocytes.

The ability of skeletal muscle to proliferate and differentiate affects the quantity and quality of muscle cells, which are important for an animal’s weight gain. GPX2 protects cells from oxidative stress. Oxidative damage can inhibit cell proliferation by promoting the expression of P53 and P21 genes in human kidney epithelial cells and pancreatic islet cells [30,31]. In addition, the inhibition of antioxidant capacity by decreasing glutathione activity impairs the regenerative capacity of muscle-derived stem cells [32]. This study also found that overexpression of GPX2, means that the oxidative stress of cells is relieved, the proliferation ability of muscle cells is strengthened. In recent years, increasing number of vitro studies have revealed that excess oxidative stress abrogated the enhancement of myoblast proliferation induced by hyperoxia [33]. Other researchers found that oxidative stress caused by hydrogen peroxide can lead to C2C12 muscle atrophy by decreasing markers of myogenic differentiation (MyoG and MyoD) [34]. Methylglyoxal (MG) increased oxidative stress and decreased myotube formation in C2C12 cells, and downregulated the expression of MyoD and myoietin (MyoG) [35]. Furthermore, for the GPX3 gene, which also has an antioxidant function, knockdown of GPX3 inactivated myoblasts of human muscle stem cells [36]. Our results have revealed that overexpression of GPX2 may enhance the proliferation and differentiation of C2C12 cells for the first time, which has certain guiding significance for the further exploration of the GPX2 gene function.

GPX2 gene expression is different in the individuals that have different polymorphism of GPX2. High GPX2 expression individuals also showed thinning 100 kg backfat thickness, no effect on AG, and lower FCR and RFI in Duroc pigs. This could be a useful gene marker in backfat and feed efficiency selection for Duroc pig populations. The reason for the above result may be that increased GPX2 gene could inhibit the adipocytes proliferation and differentiation, promoting the lipid degradation. Meanwhile, GPX2 could advance the proliferation and myogenic differentiation of muscle cells. After the high expression of GPX2 occurred, the nutrients in pigs were transferred to higher feed efficiency (Figure 8).

## Figures and Tables

**Figure 1 animals-12-03528-f001:**
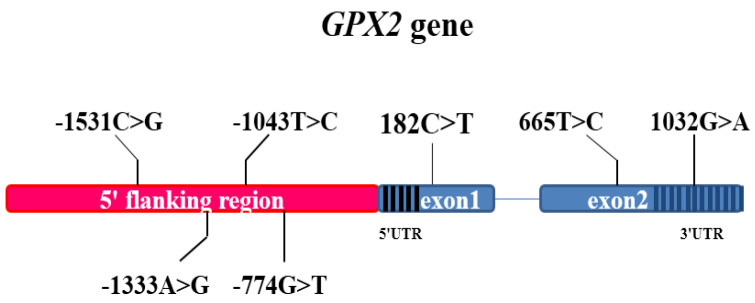
Genetic information for the pig GPX2 gene.

**Figure 2 animals-12-03528-f002:**
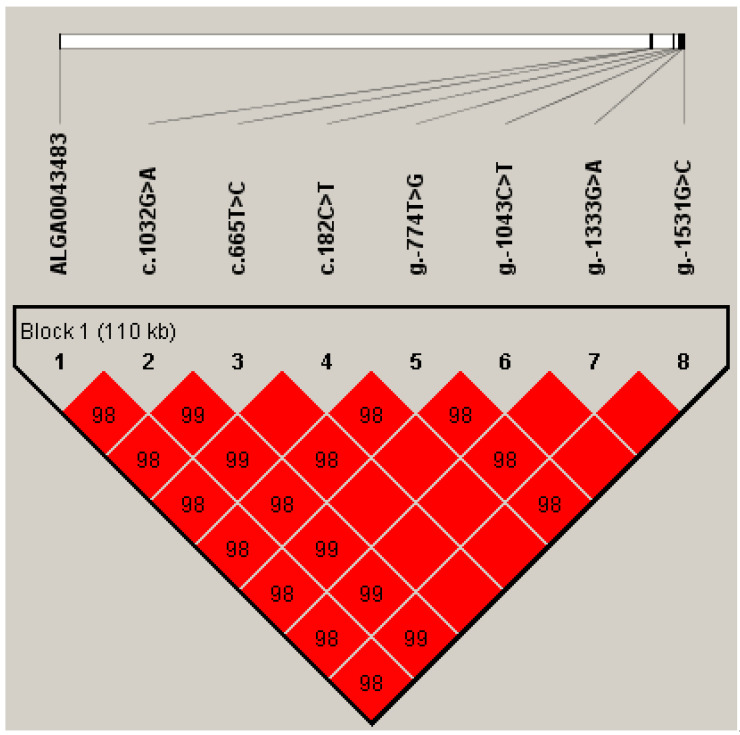
Linkage disequilibrium and haplotype block analysis using the seven SNPs for GPX2 and ALGA0043483 in 377 Duroc pigs. Solid lines mark the block identified. Linkage disequilibrium between each SNP is shown by the diamonds.

**Figure 3 animals-12-03528-f003:**
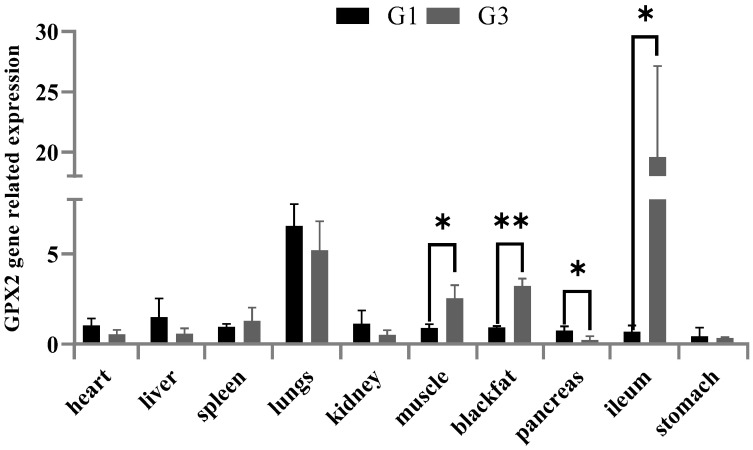
mRNA expression of GPX2 in different tissues for combination genotype G1 and G3 in Duroc pigs. combination genotype G1: GG-TT-CC-TT-CC-GG-GG, combination genotype G3: AA-CC-TT-GG-TT-AA-CC. * indicates *p* < 0.05, ** indicates *p* < 0.01.

**Figure 4 animals-12-03528-f004:**
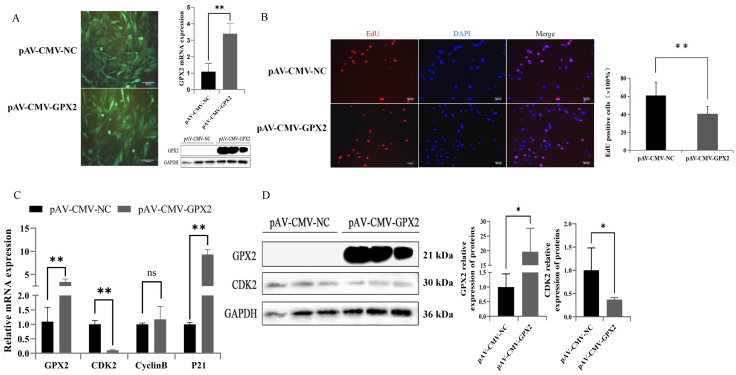
Overexpression of GPX2 inhibited the proliferation of 3T3-L1 cell. (**A**) 3T3-L1 infected with adenovirus carrying GPX2, and the mRNA and protein of GPX2 stable expressed in 3T3-L1 cell. (**B**) Representative immunostaining images comparing the cells in the proliferative phase from negative control and overexpression GPX2, the nuclei are stained with Hoechest 33342. Quantification graph showing the percentage of cells positive for the red cell (EdU). (**C**) mRNA expression of proliferation related gene between overexpression GPX2 and control group. (**D**) Protein levels and abundance analysis for GPX2 and CDK2 at 48 h after overexpression GPX2 in 3T3-L1 cell. * indicates *p* < 0.05, ** indicates *p* < 0.01, ns indicates not significant.

**Figure 5 animals-12-03528-f005:**
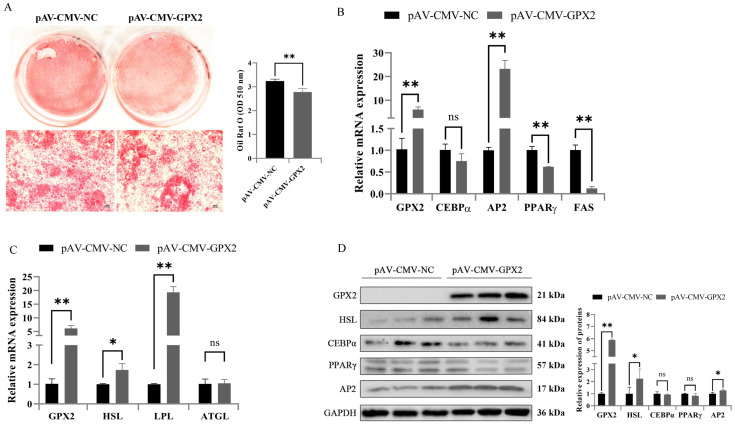
Overexpression of GPX2 inhibited adipogenic differentiation and promoted the lipid degradation of 3T3-L1. (**A**) Photograph and micrographs of 3T3-L1 stained with Oil red O, extraction and detection of lipid contents from negative control and overexpression of GPX2. (**B**) mRNA expression of adipogenic regulator and (**C**) lipid degradation in negative control and overexpression of GPX2 group of 3T3-L1 after 7 days’ differentiation. (**D**) Protein expressions and abundance analysis for adipogenic regulator and lipid degradation in 3T3-L1 after 7 days’ differentiation from negative control and overexpression of GPX2. * indicates *p* < 0.05, ** indicates *p* < 0.01, ns indicates not significant.

**Figure 6 animals-12-03528-f006:**
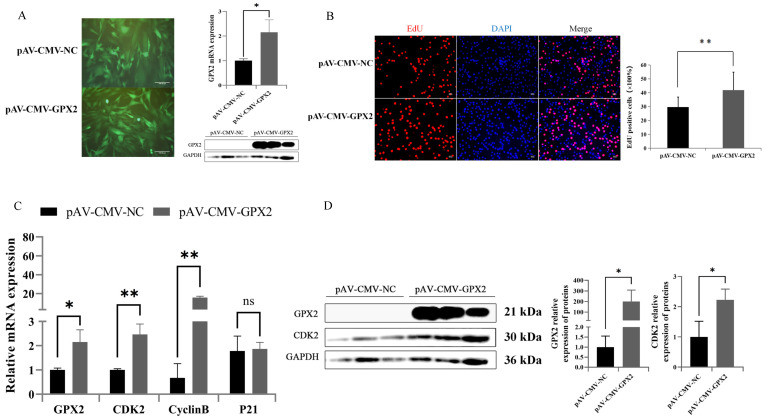
Overexpression of GPX2 inhibited the proliferation of C2C12 cell. (**A**) C2C12 infected with adenovirus carrying GPX2, and the mRNA and protein of GPX2 stable expressed in the C2C12 cell. (**B**) Representative immunostaining images comparing the cells in the proliferative phase from negative control and overexpression of GPX2, the nuclei are stained with Hoechest 33342. Quantification graph showing the percentage of cells positive for the red cell (EdU). (**C**) mRNA expression of proliferation related gene between overexpression of GPX2 and control group. (**D**) Protein levels for GPX2 and CDK2 at 48 h after overexpression of GPX2 in the C2C12 cell. In addition, abundance analysis was compared with the GAPDH protein. * indicates *p* < 0.05, ** indicates *p* < 0.01, ns indicates not significant.

**Figure 7 animals-12-03528-f007:**
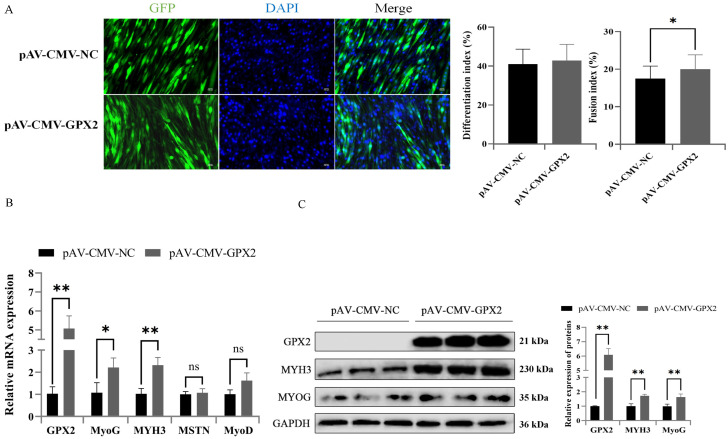
Overexpression of GPX2 promoted the C2C12 myogenic differentiation. (**A**) Representative immunostaining images comparing the cells in the myogenic differentiation stage from negative control and overexpression of GPX2. Blue represents nuclear staining; Green represents green fluorescent protein. The differentiation index was calculated as the percentage of the number of nuclei in the myotube to the total nucleus. The fusion index was calculated as the percentage of the number of myotubes with more than two nuclei in the total nucleus. (**B**) The mRNA and protein expression. (**C**) level of GPX2 and myogenic marker protein were analyzed at 7-day differentiation after overexpression of GPX2 in the C2C12 cell. * indicates *p* < 0.05, ** indicates *p* < 0.01, ns indicates not significant.

**Figure 8 animals-12-03528-f008:**
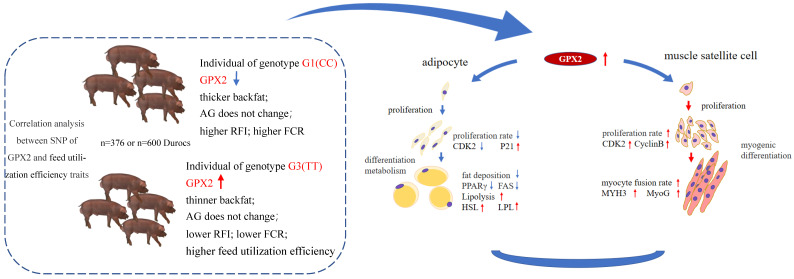
Hypothesis schematic models illustrating the GPX2 affects the backfat thickness and feed utilization efficiency of pigs by regulating the development of adipocytes and muscle cells. The left dashed box shows the correlation between SNP of GPX2 and feed utilization efficiency of Duroc pigs (*n* = 376 or 600). The individual genotype G3(TT) of GPX2 has the thinner 100 kg backfat, lower residual feed intake, lower feed convers than G1(CC) individuals. The reason for this conclusion is that the GPX2 gene could inhibit the proliferation and differentiation of adipocytes, promote the degradation of fat (mechanism diagram on the right) and advance the proliferation and differentiation of muscle cells. After the high expression of GPX2, the nutrients in pigs were transferred to higher feed utilization efficiency. ↑ Indicates expression up-regulation, ↓ Indicates expression down-regulation.

**Table 1 animals-12-03528-t001:** Genotyping assay for ALGA0043483 and seven SNPs for the GPX2 gene.

SNP	Variant Location in Ssc7	Target Region	Mutation Type	Ensemble Variant ID
c.1032 G > A	89007403	3′-untranslated region	3′-untranslated region	rs328629536
c.665 T > C	89007770	Exon2-1	Synonymous	rs318359750
c.182 C > T	89011439	Exon1	Synonymous	rs81218082
g.-774 T > G	89012396	5′flanking region 2	5′flanking region	rs339832589
g.-1043C > T	89012647	5′flanking region 2	5′flanking region	rs344114554
g.-1333G > A	89012937	5′flanking region 1	5′flanking region	rs318634907
g.-1531G > C	89013205	5′flanking region 1	5′flanking region	rs332828021
ALGA0043483 T > C	Near with GPX2 ^1^			rs80875831

The variant location was marked on the basis of Sus scrofa 11.1. ^1^ This variant had been detected by Illumina_PorcineSNP60, Axiom Genotyping Array, GGP Porcine HD. The SNP of ALGA0043483 was located in 95,283,610 bp on Ssc7 (Sus scrofa 10.2) that was near GPX2. In addition, this variant was not mapped in Sus scrofa 11.1 accurately.

**Table 2 animals-12-03528-t002:** Genotypes, allele frequencies and genetic diversity for ALGA0043483 and the seven SNP sites for GPX2, which were genotyped from 383 Duroc pigs.

SNP	Num	Genotype Frequency ∗ 100%			Allele Frequency ∗ 100%		He ^a^	Ne ^b^	PIC ^c^	HWE ^d^
c.1032G > A	381	GG	GA	AA	G	A	0.50	1.98	0.37	1.727
		18.63(71)	53.28(200)	28.08(106)	45.28	54.72				
c.665T > C	383	TT	TC	CC	T	C	0.50	1.98	0.37	2.156
		18.54(71)	53.26(203)	28.20(107)	45.17	54.83				
c.182C > T	383	CC	TC	TT	C	T	0.50	1.98	0.37	2.013
		18.54(71)	53.26(203)	28.20(107)	45.17	54.83				
g.-774T > G	383	TT	TG	GG	T	G	0.50	1.99	0.37	2.171
		19.06(73)	53.52(204)	27.42(104)	45.82	54.18				
g.-1043C > T	383	CC	CT	TT	C	T	0.50	1.99	0.37	1.902
		19.06(73)	53.00(204)	27.94(106)	45.56	54.44				
g.-1333G > A	382	GG	GA	AA	G	A	0.50	1.98	0.37	1.997
		18.59(72)	53.40(204)	28.01(106)	45.29	54.71				
g.-1531G > C	382	GG	GC	CC	G	C	0.50	1.98	0.37	2.013
		18.59(71)	53.40(204)	28.01(107)	45.29	54.71				
ALGA0043483	600	CC	TC	TT	C	T	0.50	2.00	0.38	2.202
		22.33(134)	53.00(318)	24.67(148)	48.83	51.17				

^a^ He: gene heterozygosity. ^b^ Ne: effective number of alleles. ^c^ PIC: polymorphism information content. ^d^ HWE: Hardy–Weinberg equilibrium value (**χ_0.05_^2^ =** 5.991, **χ_0.01_^2^ =** 9.21).

**Table 3 animals-12-03528-t003:** Genotype frequencies of ALGA0043483 and the seven SNPs in 377 Duroc pigs.

Haplotype	Genotype Sequence	Num	Frequency(%)
G1	CC-GG-TT-CC-TT-CC-GG-GG	70	18.47
G2	CT-GA-TC-CT-TG-CT-GA-GC	200	53.03
G3	TT-AA-CC-TT-GG-TT-AA-CC	104	27.71
G4	CC-GG-TC-CT-TG-CT-GA-GC	1	0.26
G5	TT-AA-CC-TT-TG-TT-AA-CC	2	0.53

Combination genotype 1: G1; combination genotype 2: G2; combination genotype 3: G3; combination genotype 4: G4; combination genotype 5: G5.

**Table 4 animals-12-03528-t004:** Association of each genotype of GPX2 with different feed efficiency traits.

Trait	Genotype	Value (Mean ± SE) ^1^
Birth weight (kg)	G1	1.74 ± 0.04
	G2	1.70 ± 0.05
	G3	1.72 ± 0.04
Weaning weight (kg)	G1	6.94 ± 0.30
	G2	7.26 ± 0.27
	G3	7.48 ± 0.30
90d BW (kg)	G1	29.70 ± 0.80
	G2	29.16 ± 0.61
	G3	29.27 ± 0.71
ADFI (kg)	G1	1.80 ± 0.03
	G2	1.83 ± 0.03
	G3	1.79 ± 0.03
AG (kg)	G1	0.68 ± 0.02
	G2	0.71 ± 0.01
	G3	0.69 ± 0.02
30 kg age (d)	G1	90.45 ± 1.24
	G2	91.30 ± 0.95
	G3	91.14 ± 1.11
100 kg age (d)	G1	193.96 ± 2.33
	G2	192.82 ± 1.76
	G3	193.88 ± 2.05
100 kg BF (mm)	G1	7.63 ± 0.23 ^b^
	G2	7.27 ± 0.18 ^ab^
	G3	7.16 ± 0.21 ^a^
FCR	G1	2.70 ± 0.06
	G2	2.62 ± 0.04
	G3	2.61 ± 0.05
RFI (g)	G1	3.41 ± 19.33
	G2	9.37 ± 14.32
	G3	−10.46 ± 16.72

Combination genotype G1: GG-TT-CC-TT-CC-GG-GG (*n* = 70), combination genotype G2: GA-TC-CT-TG-CT-GA-GC (*n* = 201), combination genotype G3: AA-CC-TT-GG-TT-AA-CC (*n* = 105). ^1^ The value of each trait was expressed as the mean ± standard error. (a, b) mean *p* < 0.1.

**Table 5 animals-12-03528-t005:** Association of ALGA0043483 of GPX2 with different feed efficiency traits in 600 Duroc pigs.

Trait	Genotypes	Value (Mean ± SE) ^1^
Birth weight (kg)	CC	1.85 ± 0.043
	TC	1.80 ± 0.038
	TT	1.83 ± 0.042
Weaning weight (kg)	CC	8.12 ± 0.24 ^a^
	TC	8.18 ± 0.21 ^a^
	TT	8.51 ± 0.23 ^b^
90 d BW (kg)	CC	30.19 ± 0.71
	TC	29.89 ± 0.62
	TT	30.42 ± 0.70
ADFI (kg)	CC	1.70 ± 0.04
	TC	1.70 ± 0.03
	TT	1.69 ± 0.04
AG (kg)	CC	0.66 ± 0.02
	TC	0.67 ± 0.01
	TT	0.66 ± 0.02
30 kg age (d)	CC	89.71 ± 1.09
	TC	90.174 ± 0.96
	TT	89.35 ± 1.08
100 kg age (d)	CC	192.34 ± 1.86
	TC	193.57 ± 1.57
	TT	192.80 ± 1.84
100 kg BF (mm)	CC	7.87 ± 0.18 ^A^
	TC	7.88 ± 0.16 ^A^
	TT	7.46 ± 0.18 ^B^
FCR	CC	2.62 ± 0.05
	TC	2.61 ± 0.04
	TT	2.59 ± 0.05
RFI (g)	CC	−8.56 ± 21.36
	TC	−3.65 ± 18.52
	TT	−18.58 ± 21.08

CC (*n* = 134), CT (*n* = 318), TT (*n* = 148). ^1^ The value of each trait was expressed as the mean ± standard error. (a, b) mean *p* < 0.05, (A, B) mean *p* < 0.01.

## Data Availability

Not applicable.

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
