# Peer review of "GPX2 Gene Affects Feed Efficiency of Pigs by Inhibiting Fat Deposition and Promoting Muscle Development"

_animals, 2022, doi:10.3390/ani12243528_

Round 1

Reviewer 1 Report

The paper tried to dissect the association between GPX2 gene and feed efficiency and fat deposition traits. The rational of the study is relevant and of interest to the pork industry. However, the draft needs major revisions before being considered for publication. First, the author tried to do too much without clear reasons or presentation of the methodology used. In fact, the authors use general terminology that is often imprecise and sometimes not correct. The draft will benefit greatly if the authors consult with a statistician. Both equations (lines 73 and 224) are incorrect. Furthermore, the authors indicated that they used a mixed model where the effects of genotypes were assumed random. However, they failed to provide the distributional form assumed for the random effects. Additionally, it not clear if they used only the SNP with the GPX2 gene or all the SNPs in the array.

In the experimental design, the data was collected at different ages and weights of the animals without providing a clear rational or discussing the implications on the results.

The draft is very poorly written, and it requires a thorough revision and editing by an English speaker. I understand that the authors are not English speakers, but it was really hard for me to read the draft.  

Consequently, I suggest that the draft be revised, and the major modifications carried out before further consideration for review.

Author Response

The first reviewer's comments and the author's revised reply

The paper tried to dissect the association between GPX2 gene and feed efficiency and fat deposition traits. The rational of the study is relevant and interest to the pork industry. However, the draft needs major revisions before being considered for publication. First, the author tried to do so much without clearing reasons of presentation of the used methodology. In fact, the authors use general terminology that is often imprecise and sometimes not correct. The draft will benefit greatly if the authors consult with a statistician.

It is true that the English writing of this article is not 100 percent correct. So, I am deeply sorry for it. We have asked native English speakers to revise the article. This paper is only a preliminary study, not systematic, even there are many mistakes. I will also make a lot of efforts to revise the article. Please continue to give your valuable advices.

First question:Both equations (lines 73 and 224) are incorrect. Furthermore, the authors indicated that they used a mixed model where the effects of genotypes were assumed random. However, they failed to provide the distributional form assumed for the random effects. Additionally, it not clear if they used only the SNP with the GPX2 gene or all the SNPs in the array.

First response:Thank you very much for your comment. We have made the following changes in lines 80 and 233-243: The conversion abnormality of the formula in line 73 was not shown when the formula input function was used, and the author did not find this error in the manuscript check. I'm sorry for my carelessness. Here, is the added formula to the text and it is marked them with red.

The formula line 80, it has been carefully modified as follows:

RFI = DFI - [b1 x onBW + b2 x offBW + b3 x metamidBW + b4 x ADGA + b5 x offBFA + e]

The formula line 233, after careful examination, the formula is indeed not rigorous, and there are mistakes. It has been carefully modified as follows:

Y = μ + G + S + P + A + W + e

Where Y is the observation of the trait (ADFI, ADG, 100kg age, 100kgBF, FCR and RFI), μ is the population mean, G is the fixed effect of the combination genotype, S is the random effect of sex, P is the random effect of the batch of test, A is the random effect of parity, W is the covariate effect of onBW, e is the random error.

Where Y is the observation of the trait (Birth weight, Weaning Weight, 90dBW, 30kg age), μ is the population mean, G is the fixed effect of the combination genotype, S is the random effect of sex, didn’t include P and A as the random effect, W is the covariate effect of Birth Weight (When Y was Weaning Weight), W is the covariate effect of Weaning Weight (When Y was 90dBW, 30kg age), When we analyze Y as the birth weight, no covariable was set, e is the random error.

The regression equations used for correlation analysis of different traits are not the same here. All association analyses used combined (all) SNP results.

Second question:In the experimental design, the data was collected at different ages and weights of the animals without providing a clear rational or discussing the implications on the results.

Second response:In this experiment, growth performance-related traits were all used in pigs around 90 days of age. Until the end of the detection period, about 190 days of age. In addition, all growth performances were performed using accurate whole-course data from 90 to 180 days without age indication. These data are strictly controlled. Weaning weight and birth weight are the actual data collected. The author also did a lot of work in the process of analyzing the data.

Third question:The draft is very poorly written, and it requires a thorough revision and editing by an English speaker. I understand that the authors are not English speakers, but it was really hard for me to read the draft.

Third response:Thanks a lot for your comment. I have entrusted the draft to English speakers to polish it. If there are any inappropriate things, your suggestions and advices are welcome.

Fourth question:Consequently, I suggest that the draft be revised, and the major modifications carried out before further consideration for review.

Fourth response:Thank you very much for your advice. I have made significant revisions to the draft and it is marked in red. But I will make a further revisions on this draft to make it good.

Reviewer 2 Report

Dear authors, the manuscript fits the Journal scope and has relevant and original content. However, I noticed some editing errors that can be fixed. In addition, I have suggested some sentence changes to make it easier for readers to interpret.

Page 1, Line 42: Please add the reference for the sentence “Ob/ob mice showed a decrease in adipocyte cellular activity of GPXs”.

Page 2, Line 51-52: I suggest for authors to change the sentence “The present study detected sequence variants of GPX2 in duroc pigs (Sus scrofa domestics) and investigated their associations with feed efficiency traits” to “The present study was carried out to detect sequence variants of GPX2 in duroc pigs (Sus scrofa domestics) and investigate their associations with feed efficiency traits”.

Page 2, Line 71: Were the males castrated? If yes, please replace “boar” by “barrows”. I think the “gilts” is more appropriate to describe females pigs at this age instead of “sows”.

Page 3, Line 121: For some reason, the word “adipogenic” is written in smaller letter size.

Page 3, Line 123: Please, describe DMEM before the abbreviation.

Page 3, Line 122-127: Please, change the verb tense of the sentence.

Page 4, Line 151-153: Please, change the verb tense of the sentence.

Page 4, Line 155-161: Please, change the verb tense of the sentence.

Page 4, Line 168-171: Please, change the verb tense of the sentence.

Page 4, Line 182: Please, replace “can be” by “was”.

Page 5, Line 196-199; 207: Please, change the verb tense of the sentences:

Page 6, Line 251: Please, replace “heterozygosity” and “number of alleles” by its abbreviations, “He” and “Ne”, respectively.

Page 18: Figure 5 is repeated.

Page 22, Line 414: Please, cite the studies that confirmed your results.

Page 22, Line 417: Does “75-117 cm” mean “75-117 cM”?

Page 24, Line 434: What do you mean by “more efficient diet”? I suggest to replace “more efficient diet” by “more feed efficiency”.

Page 24, Line 449: Does “TR treatment” mean “Tyrozol”? If yes, please insert the abbreviation (TR) after Tyrozol”.

Page 24, Line 451-458: Please, describe the abbreviations, HFD and WAT. 

Page 25, Line 489-491: The last sentence is confusing, if possible, rewrite it.

Author Response

The second reviewer's comments and the author's revised reply

Dear authors, the manuscript fits the Journal scope and has relevant and original content. However, I noticed some editing errors that can be fixed. In addition, I have suggested some sentence changes to make it easier for readers to interpret.

First question:Page 1, Line 42: Please add the reference for the sentence "Ob/ob mice showed a decrease in adipocyte cellular activity of GPXs".

First response:Thank for your comment. We have made the following changes in lines 43: The reference of the sentence "Ob/ob mice showed a decrease in adipocyte cellular activity of GPXs" is [7]. I'm sorry for my carelessness. Here, the reference is added to the text and marked in red.

Second question:Page 2, Line 51-52: I suggest for authors to change the sentence "The present study detected sequence variants of GPX2 in duroc pigs (Sus scrofa domestics) and investigated their associations with feed efficiency traits" to "The present study was carried out to detect sequence variants of GPX2 in duroc pigs (Sus scrofa domestics) and investigate their associations with feed efficiency traits".

Second response:Thank you for your suggestion. Based on your recommendations, we have changed lines 57-58 of sentences "The present study detected sequence variants of GPX2 in duroc pigs (Sus scrofa domestics) and investigated their associations with feed efficiency traits" to "The present study was carried out to detect sequence variants of GPX2 in duroc pigs (Sus scrofa domestics) and investigate their associations with feed efficiency traits".

Third question:Page 2, Line 71: Were the males castrated? If yes, please replace "boar" by “barrows”. I think the "gilts" is more appropriate to describe female pigs at this age instead of "sows".

Third response:Thank you for your suggestion. The males are not castrated. We have made the following changes in line 77: As suggested by the reviewer, we have replaced "sows" with "gilts". I modified in the text and marked it in red.

Fourth question:Page 3, Line 121: For some reason, the word "adipogenic" is written in smaller letter size.

Fourth response:Thank you for your suggestion. I'm sorry for my carelessness. We have made the following changes in line 128: The word "adipogenic" is written in small letter size. I have modified it in the text and marked it in red.

Fifth question:Page 3, Line 123: Please, describe DMEM before the abbreviation.

Fifth response:Thank you for your comments. We have made the following change in line 130: DMEM means: Dulbecco minimum essential medium. I have described it in the text and marked in red.

Sixth question:Page 3, Line 122-127: Please, change the verb tense of the sentence.

Sixth response:Thank you for your comments. We have made the following changes in line 129-135: I have changed the verb tense of the sentence in the text.

Seventh question:Page 4, Line 151-153: Please, change the verb tense of the sentence.

Seventh response:Thank you for your comments. We have made the following changes in line 159-161: I have changed the verb tense of the sentence in the text and marked them in red.

Eighth question:Page 4, Line 155-161: Please, change the verb tense of the sentence.

Eighth response:Thank you for your comments. We have made the following changes in line 163-171: I have changed the verb tense of the sentence in the text and marked them in red.

Ninth question:Page 4, Line 168-171: Please, change the verb tense of the sentence.

Ninth response:Thank you for your comments. We have made the following changes in line 177-181: I have changed the verb tense of the sentence in the text and marked them in red.

Tenth question:Page 4, Line 182: Please, replace "can be" by "was".

Tenth response:Thank you for your suggestion. We have made the following changes in line 191: I have changed "can be" by "was" in the text and it is marked in red.

Eleventh question:Page 5, Line 196-199; 207: Please, change the verb tense of the sentences.

Eleventh response:Thank you for your comments. We have made the following changes in line 205-207 and 209: I have changed the verbs tenses of the sentences in the text and marked them in red.

Twelfth question:Page 6, Line 251: Please, replace "heterozygosity" and "number of alleles" by its abbreviations, "He" and "Ne", respectively.

Twelfth response:Thank you for your suggestion. We have made the following changes in line 264: We have replaced "heterozygosity" and "number of alleles" by "He" and "Ne" as you suggested in the text and marked them in red.

Thirteenth question:Page 18: Figure 5 is repeated.

Thirteenth response:Thank you for the advice. I am very sorry for my carelessness that figure 5 has been reused for editorial typography. I have removed it from the text.

Fourteenth question:Page 22, Line 414: Please, cite the studies that confirmed your results.

Fourteenth response: The reviewer's questions, which are particularly professional, have addressed the core weakness of this article.

This paper also predicts in the attached table, c.1032G. The mutation at the A site greatly affects the secondary structure of GPX2 mRNA and may affect the gene activity. But to confirm this prediction, the technology and methods are not yet available to us. Structural changes that affect the activity and number of genes. Therefore, this paper promoted gene expression from the perspective of quantity, and studied the functional influence of subsequent genes.

In addition, in order to make the cite here more reasonable, we have made the following changes in line 406: Before modification: "Based on the previous studies, we speculated that a variety of secondary structures may have something to do with the function of GPX2." Modified to:"Based on the previous studies, we speculated that a variety of secondary structures may have something to do with the function of GPX2. This aspect needs further study." And marked them in red.

Fifteenth question:Page 22, Line 417: Does "75-117 cm" mean "75-117 cM"?

Fifteenth response:Thank you for the advice. We have made the following changes in line 417: I am very sorry for my carelessness that "75-117 cm" mean "75-117 cM". I have revised it in the text and marked them in red.

Sixteenth question:Page 24, Line 434: What do you mean by "more efficient diet"? I suggest to replace "more efficient diet" by "more feed efficiency".

Sixteenth response:Thank you for your professional advice. We have made the following changes in line 434: I have replaced "more efficient diet" with "more feed efficiency" in the text and marked it in red.

Seventeenth question:Page 24, Line 449: Does "TR treatment" mean "Tyrozol"? If yes, please insert the abbreviation (TR) after Tyrozol.

Seventeenth response:Thanks for your advice. We have made the following changes in line 447: "TR treatment" mean "Tyrosol". I have inserted the abbreviation (TR) after "Tyrosol" in the text and marked it in red.

Eighteenth question:Page 24, Line 451-458: Please, describe the abbreviations, HFD and WAT.

Eighteenth response:Thank you for your comment. We have made the following changes in line 451-459: HFD and WAT means: high fat diet and white adipose tissue. I have described it in the text and marked in red.

Nineteenth question:Page 25, Line 489-491: The last sentence is confusing, if possible, rewrite it.

Nineteenth response:We have made the following changes in line 487-491: Before modification: "The reason for this conclusion is because that the increased GPX2 gene could inhibit the adipocytes proliferation and differentiation, promoted the lipid degradation. Meanwhile, GPX2 could advances the proliferation and myogenic differentiation of muscle cells. After the high expression of GPX2 conducted the nutrients in pigs were transferred to higher feed efficiency (Figure 8)." Modified to:"The reason for the above result may be that elevated GPX2 gene could inhibit the adipocytes proliferation and differentiation, promoted the lipid degradation. Meanwhile, GPX2 could advances the proliferation and myogenic differentiation of muscle cells. After the high expression of GPX2 conducted the nutrients in pigs were transferred to higher feed efficiency (Figure 8)." And marked them in red.

Reviewer 3 Report

I have gone through the manuscript. The results of this study will contribute to a better understanding of how the GPX2 gene affects the feed efficiency of pigs. Still, there is a lack of explanation, and the reference should be checked throughout the article.

Line 24: What does the “special genotype of GPX2” mean?

Line 34-53: In the Introduction section, reference numbers are inconsistent. Authors should check the other section. In addition, the introduction should be better elaborated (i.e., how concerned the significant SNP to feed intake (RFI) in the previous report, whether in vitro experiments have been performed, what is the breed of the pig? and so on).

Line 60-80: ADG abbreviation used for the first time should be mentioned what it means, and is "average" needed? About the formura “RFI = DFI - [1] ” what does [1] mean? Additionally, the formula for the calculations at age 30 and 100 kg should be shown.

Line 134: differentatiatio”n”

Line 226-227: Please provide fathers and mothers information more details. Is grandfathers and grandmothers information are taken into account?

Table 5: The weaning weight significantly differed among genotypes, which was not mentioned in the results and discussion section. Did age at weaning differ among genotypes? Additionally, couldn't these difference affect the back fat's thickness and muscle increase afterward?

Line 414-422: The explanation about GPX2 in the previous report should be transferred to the "Introduction" section.

Line 428-430: The reason why daily gain had not been affected was understandable in this text but not understandable only in Figure 8. Therefore, I suggest modifying Figure 8 to make it understandable, adding daily gain. And I think it's just a hypothesis.

Line 462-464, 481-483: Further data analyses or another experiment are also needed for the suggestion. Were these inhibition or promotion dependent on GPX2 expression levels in the cultured cell? 

Author Response

I have gone through the manuscript. The results of this study will contribute to a better understanding of how the GPX2 gene affects the feed efficiency of pigs. Still, there is a lack of explanation, and the reference should be checked throughout the article.

First question:Line 24: What does the "special genotype of GPX2" mean?

First response:Thanks for your question. We have made the following changes in line 24: "special genotype of GPX2" means "G3+TT combination Genotype". It has been revised in the article and marked in red.

Second question: Line 34-53: In the introduction section, reference numbers are inconsistent. Authors should check the other section. In addition, the introduction should be better elaborated (i.e., how concerned the significant SNP to feed intake (RFI) in the previous report, whether in vitro experiments have been performed, what is the breed of the pig? and so on).

Second response:Thank you for your comments. In the introduction section in line 47, the reference number has been added and marked in red. Other parts were also checked.

Third question:Line 60-80: ADG abbreviation used for the first time should be mentioned what it means, and is "average" needed? About the formula "RFI = DFI - [1]" what does [1] mean? Additionally, the formula for the calculations at age 30 and 100 kg should be shown.

Third response:Thank you for your comments. We have made the following changes in line 75 and 80: ADG means: average daily gain. I have described it in the text, marked in red. The conversion abnormality of the formula in line 80 was not shown when the formula input function was used, and the author did not find this error in the manuscript check. I'm sorry for my carelessness. Here the formula in line 80 is added in the text and marked in red.

The formula line 80, it has been carefully modified as as follows:

RFI = DFI - [b1 x onBW + b2 x offBW + b3 x metamidBW + b4 x ADGA + b5 x offBFA + e]

30~100kg ADG, 30kg age and 100kg age. All three indexes are quoted from Canadian swine improvement program [10].

30~100kg ADG formula always contains each factor that is longer. It's a bit tedious to list them all in the article. We will list them if needed.

30~100kg ADG=(70kgx1000g/kg)/(100kg age-30kg age)

30kg age=start age of test+[(30kg-stater body weight)xb (Duroc:1.536)

100kg age=off age of test-[(off test body weight -100)/CF

CF=(off test body weight+off test age)x1.826040(boar)

CF=(off test body weight+off test age)X1.714615(Gilt sow)

Fourth question:Line 134: differentatiatio"n"

Fourth response:Thanks for the comment. We have made the following changes in line 142: I'm sorry for my carelessness by missing an "n" for the word "differentiation". I have revised it in the text and marked them in red.

Fifth question:Line 226-227: Please provide fathers and mothers information more details. Are grandfathers and grandmothers’ information are taken into account?

Fifth response:After verification, there is no use of parent generation information in GLMM, so the parent effect has been deleted, see the content of the update text please.

Sixth question:Table 5: The weaning weight significantly differed among genotypes, which was not mentioned in the results and discussion section. Did age at weaning differ among genotypes? Additionally, couldn't these differences affect the back fat's thickness and muscle increase afterward?

Sixth response:Table 5: Weaning weight data, the factory adopts 28 days of age to weigh, recorded as weaning weight, after the completion of the weighing period weaning. Therefore, the weaning weight here is 28 days of age, and the data are relatively accurate. In this paper, the growth performance was measured around the age of 90 days, but the specific date was different, because the measuring station could measure the weight of each pig every day. After the pre-feeding period, the accurate weight of 90 days of age was used as a covariate to control the influence of different measuring days of age or weight on the growth performance.

Seventh question:Line 414-422: The explanation about GPX2 in the previous report should be transferred to the "Introduction" section.

Seventh response:Thank you very much for your advice. An explanation of GPX2 has been added in the second paragraph of the "Introduction" section in line 47-53 and is marked in red. The additions are: “In addition, quantitative trait loci (QTL) in Porcine chromosome 7 was found to correlate with back fat on 3-4th lumbar vertebrae, 11-12th rib and neck fat thickness [8]. Other similar reports have found the back fat traits was affected by the QTL (region: 75-117 cM on porcine chromosome 7) which contained GPX2 [9]. In addition, several studies also have shown that GPX2 gene was involved in lipid metabolism [10]. Therefore, GPX2 is considered to be an important candidate gene for feed utilization efficiency and porcine fat deposition."

Eighth question:Line 428-430: The reason why daily gain had not been affected was understandable in this text but not understandable only in Figure 8. Therefore, I suggest modifying Figure 8 to make it understandable, adding daily gain. And I think it's just a hypothesis.

Eighth response:We have made the following changes in line 430: The Figure 8 has been revised according to your opinions, add the ADG didn’t change. As the experts said, this schematic model is indeed just a hypothesis, which needs to be confirmed by many experiments in vivo. Therefore, in the section of schematic models, we modified the title as hypothesis schematic models to emphasize the meaning of hypothesis.

Ninth question:Line 462-464, 481-483: Further data analyses or another experiment are also needed for the suggestion. Were this inhibition or promotion dependent on GPX2 expression levels in the cultured cell?

Response:Thanks for the question. As you said, there are still needed many systematic studies to well illustrate the function of GPX2 on backfat thickness and feed utilization efficiency. And that's what we're going to do in the future.

And in this paper, we conducted experiments at the cellular level after overexpression of GPX2. The difference between the experimental group and the control group, which can explain the promotion or inhibition the growth of cell caused by the increased expression of GPX2, to some extent.
